# Government Intervention on Cooperative Development in Poor Areas of Rural China: A Case Study of XM Beekeeping Cooperative in Sichuan

**Shemei Zhang [1], Bin Wu [2], Rui Chen [1], Jingzhi Liang [3], Nawab Khan [1,*] and Ram L. Ray [4]**

1    College of Management, Sichuan Agricultural University, Chengdu 611100, China
2    School of Business, Nottingham University, Nottingham NG8 1BB, UK; bin.wu@nottingham.ac.uk
3    Party School of Dujiangyan Municipal Committee of the CPC, Chengdu 611844, China
4    College of Agriculture and Human Sciences, Prairie View A&M University, Prairie View, TX 77446, USA
*    Correspondence: nawabkhan@stu.sicau.edu.cn

**Abstract:** The relationship between government intervention and cooperative development has always been a source of controversy in the developing world. This paper aims to examine the rationale and successful conditions of government intervention to promote cooperative development in poor areas of rural China. In the context of the "targeted poverty alleviation" program (2015–2020), a government-led campaign covering all poverty-stricken villages in west China, cooperative development was listed by the central government as a criterion for evaluating successful intervention at the county government level. Accordingly, the central questions of this paper are: why is government intervention necessary to initiate a process of cooperative development in poor areas of China; and under what conditions can government intervention be successful, leading to sustainable cooperative development? Bearing in mind the complexity of government intervention with mixed results, both successful and failed, the above questions are addressed through a case study of XM Beekeeping Cooperative, representing one type of successful government intervention in poverty-stricken and ethnic-minority-dominated regions of China. Overall, government intervention is crucial in building cooperative ecosystems in poor regions of China. However, government intervention is not invariable because the approaches can be modified to accommodate the effect of the intervention.

**Keywords:** governmental intervention; cooperative development; successful intervention; poor areas of rural China

## 1. Introduction

For rural development in the developing world, it is recognized that cooperatives can play a positive role in helping smallholder farmers by creating economic opportunities, accessing markets and key resources, increasing bargaining power, and reducing individual risk [1]. In the context of poverty alleviation in poor areas of the Global South, the first UN Sustainable Development Goal, a puzzle facing the international community is to understand the role of government intervention in cooperative development, which may conflict with the cooperative principles taken by the International Cooperative Alliance. In this regard, China is the largest developing country in the world, and its experience in cooperative development is worth examining in detail, given its strong government intervention in rural development including cooperative development in its poor areas.

The value of considering China in debates concerning government intervention and cooperative development in the Global South is related to the fact that the Chinese government has made a serious effort to foster cooperative development nationwide since 2007 to empower its 230 million smallholder farmers (nearly half of the world's smallholder farmers in total) who are dispersed, poor, and vulnerable regarding their bargaining power in agricultural supply chains. Equally important is its national campaign, namely the

"targeted poverty alleviation" program (TPA, 2015–2020) in remote, mountainous, ethnic-minority populated, and poverty-stricken areas. Cooperative development has been listed as a key criterion for the success of government intervention at the county level. Through national mobilization and coordination of all types of resources (e.g., technological, physical, financial, and talent-related) from all types of organizations (e.g., governmental agencies, universities, research institutes, enterprises, and non-profit organizations) and regions (including coast provinces and the largest municipal areas), it is undoubtedly that such a strong intervention would remove various constraints (e.g., geographic, resource-based, and infrastructure-related) affecting local economic development effectively, which has provided new momentum for cooperative development in China's poor areas. According to official figures, government intervention has resulted in not only the decline of the population of those in poverty from 98.99 million in 2012 to 5.51 million in 2019 but also the rapid growth of "farmers" professional cooperatives, which reached 685,000 in poor areas in total, covering over 90% of poor villages, benefitting 21.98 million rural people.

In theory, strong government intervention in cooperative development in the developing world, including rural China, conflicts with the Cooperative Values and Principles adopted by the International Cooperative Alliance (ICA) and the international community. In practice, top-down intervention is not always successful or effective, and there are many reports concerned about "faded cooperatives" or suspicions about their sustainability [2]. Two questions arise: Why is governmental intervention necessary to initiate a process of cooperative development in poor areas of rural China? and under what conditions can external intervention be successful, leading to sustainable cooperative development? The above questions may be challenging to address considering the complexity of rural development and government intervention in poor areas of China and the interconnection of many factors between geography, resources, climate, infrastructure, and economic, social, andsocial–cultural conditions. As part of a GCRF-funded project focusing on cooperative ecosystems to empower small farmers in poor areas of China [3], a case study of a Xing-Mu beekeeping cooperative (referred to as XM Cooperative hereafter) has been conducted in HS county, a high, mountainous, and Tibet-minority County of Sichuan, to provide insight into the rationale and successful (or failed) conditions of government intervention for rural development, including cooperative development, in poor areas of China.

The article is organized into seven sections. The Section 2 reviews the debates around governmental intervention and cooperative development in the context of the developing world. It is followed by a background of China's cooperative movement and governmental intervention. Section 4 discusses the methodology of our case study and field research. It is followed by a narrative of the XM Cooperative case in Section 5 and discussion of research findings in Section 6. This paper ends with a conclusion in Section 7.

## 2. Literature Review

Since the establishment of the first cooperative (the Rochdale Society of Equitable Pioneers) in England in 1844, principles of cooperatives have been widely adopted or tested not only because they provide "a much more democratic system for people used to autocratic domination", but also because they offer an alternative model of "capitalist globalization" [4]. Compared to the community and volunteer-based model in the developed world, cooperative development in the developing world is more complicated and controversial concerning the role of government intervention. The term cooperative development in this paper is broadly referred to as two interconnected processes: creating and growing one cooperative and developing a favorable and robust cooperative environment. This definition is different from the narrow description of cooperative development to "catch both the effort that goes into developing the larger cooperative environment, as well as the required connection between the new co-op and the larger cultures within which co-ops operate" [5]. Cooperative development in the developing world must understand the context of cooperative development. This is complicated, multi-layered, and multi-faceted [5]. The variety of "cooperative development contexts" calls for attention to

"critical fit" [6] because "what works smoothly within the culture of Bangladesh may not quite fit in Indonesia or central India, or elsewhere". "Critical fit" requires a match between the goals of the cooperative and the goals of the community because a successful model of cooperative development at one location may fail in another, in terms of cooperation, due to different cultures [4].

Government intervention as one of three "lead actors" (the other two are: existing or new cooperatives and cooperative developers) for cooperative development [5] can play different roles depending upon the "cooperative development context". With a focus on the initiative or support of cooperative development in the developing world, for instance, three approaches can be identified: (1) the government can act as a self-selecting group formed by people with prior social cohesion (neighbors, friends, and acquaintances) to address a need; (2) the government can act as a constructed group created at the initiative of an external party; and (3) the government can act as a replica- or model-driven group built by using a successful cooperative prototype [5]. Unlike the first approach, government agencies in the developing world can play a positive role in the last two approaches in stimulating local participation in cooperative development and can provide various services, including researching the demands and needs of the community; educating the community about the importance of cooperation; and linking participation directly with member's livelihoods, among many others [4].

Among the many roles the government may play, the most significant one is that of providing a financial environment and support for cooperative development, which varies significantly from country to country. In contrast to the self-regulating micro-finance system created in Bangladesh (Grameen model), for instance, the Indonesian government played a strong role in encouraging and capitalizing on cooperatives and has been responsible for many cooperatives that became "heavily dependent on the government to maintain their programs" [4]. Despite the conflict with the principle that a cooperative should remain as independent of the government as possible, the Indonesian case of strong governmental intervention is not unique but shared with many countries in the developing world. Regardless of their different orientations, according to Develtere et al. [7], all of the cooperatives in Africa by the early 1990s shared a common feature as dependent agents or clients of the state and other semi-public agencies, which hardly operated as private business enterprises and were driven by the interests of their members. Furthermore, there are two popular but contradictory views on government intervention: the failure or malfunctioning of cooperatives caused by the hijacking of governments and nurturing of cooperative entrepreneurship for African development (ibid: xiii).

Viewing strong government intervention as a major cause for the underdevelopment of the African continent, the structural adjustment under the neo-liberalism approach since the early 1990s has given favor to market mechanisms and voluntary initiatives. An academic bias, according to Develtere et al. [7], appears among organizations and researchers who have "very little interest or attention" paid to cooperative development because the latter "more or less serviced the interests of the state than the ordinary members and the general public". Like the African experience, the European cooperative model was popular in Latin America after World War II through a top-down governmental intervention designed to address challenging issues over inequality and widespread poverty in general and promote production cooperatives in more marginal rural areas [8]. As CEPAL [9] concluded from their study of cooperatives in Colombia, Venezuela, and Ecuador, "the cooperative movement was imposed from the top as a paternalistic and authoritarian act; it was not the result of popular conviction based on democratic participation or popular enlightenment."

Bringing together cooperative development practiced throughout Latin America and Africa, Gagnon [10] emphasized that cooperatives in capitalist societies abandoned their role as social movements to become better integrated into the dominant capitalist system while cooperatives in socialist countries became little more than vehicles for the transmission of state policy. Ghosh [11] attributes the failure of cooperative development in India to government interference in a rapid increase in cooperative membership without attention

paid to the lack of a cooperative spirit among rural people from the beginning. However, he notes the necessity of government involvement in terms of technical contributions, management inputs, and some finance at the initial stages of formation. Like India's case, China had an experience unsuccessful governmental intervention in cooperative development in theearly–mid 1950s, leading to a failed collective commune system for two decades until the early 1980s. The new wave of the cooperative movement in rural China since 2007 has triggered intellectual debates along the line of rural sustainability and the possibility of China pursuing a third way of development beyond capitalism and collectivism [2,12]. With a geographic focus on poor areas of China, nonetheless, many researchers show positive outcomes of governmental intervention for poverty alleviation and self-organization among small farmers through a lens of e-commerce ecosystems [13,14].

There are two models of cooperative development in the developing world: a bottom-up model (e.g., the micro-finance system in Bangladesh) and a top-down model (e.g., the Indonesian case). The two models do not necessarily conflict, but neither are they interchangeable. Taking the case of the government-sponsored banking system in Indonesia as an example, it has become a cooperative system for financing small independent enterprises and cooperatives entirely. The Indonesian case raises questions about how much a cooperative enterprise should be open to government influence.

Cooksey and Kikula [15] found a wide implementation of top-down, bottom-up, and mixed approaches in Tanzania. They further discussed that the top-down approach had dominated the planning cycles for a long time in Tanzania and many other parts of the world. One of the main reasons for this dominance is that it allows rapid large-scale spending of budgets in accordance with pre-established timetables. However, Zwane and Kekana [16] recommended the bottom-up approach over the top-down approach in Southern Africa for agriculture cooperatives because the latter approach left no room for dependency. In addition to top-down and bottom-up approaches, Casazza and Pianigiani [17] found a commercial approach, which can also be implemented in urban agriculture cooperatives. Gezahegn et al. [18] investigated the performance of agricultural cooperatives in Ethiopia. They found that the cooperatives initiated in a bottom-up approach were more efficient than those following the top-down approach initiated through the government and NGOs.

For some fear that the possibility of easy financing from government or corporate sources will dilute the cooperative spirit, Williams [4] suggests that the genius of the cooperative movement is the strength of its grass-roots support so that government intervention "makes little difference in the long run" because many co-ops thrive anyway and become independent of outside support.

## 3. Background for Government Intervention in "Farmers" Cooperative in Rural China

The necessity of research focusing on government intervention can be seen from the uneven experience of the cooperative movement in China in the past seven decades. With a vision of the modernization of the agriculture and industry system in China, the transformation of smallholder farmers via "producers" cooperation organizations has been listed as a key element for rural development since the Communist Party took national power in 1949. Government intervention, however, has taken different paths and formatting in the last seven decades. This can be divided into four broad periods with different features: farmers' self-organization with government support (1949–1956); a collectivism and commune system with government control (1957–1978); a de-collectivism reform and household responsibility system (1979–2007); and new momentum for cooperative movement since the validity of the Farmers' Professional Cooperatives Law in 2007.

The rural reform beginning in 1978 resulted in a two-tier system consisting of household contract management and a collective economy. On the premise of keeping the joint ownership of rural land unchanged, rural householders have equally accessed and managed the collective land as independent producers for the external market. In contrast, collective economic organizations were designed to develop their functions to provide

various public services and coordination between individual householders in production and sale. While rural reform promoted the development of the market economy, the free flow of rural labor, and the competition among smallholders, it has failed to develop second-tier, collective economic organizations to provide public services and coordinate for individual farmers. In addition, the fragmentation of arable land resulting from the HRS makes it very difficult to achieve an economy of scale, mechanization, and the adoption of new technologies.

In 2007, the Chinese government formally implemented the Law on Farmer Specialized Cooperatives (FPC) to standardize, encourage, support, and guide the development of farmer cooperatives and protect the legitimate rights and interests of the cooperatives and their members. Since then, cooperative development in rural China has experienced rapid growth, which has provided new momentum for rural development concerning various aspects, including land circulation, the emergence of professional managers of cooperatives, new mechanisms for cooperative management, and collaboration with community and external enterprises. The scope of farmer cooperatives has not only covered a range of production areas (e.g., grain, cotton, oil, meat, eggs, milk, fruits, vegetables, tea, and other major products), but has extended to agricultural product processing, rural tourism, folk craft production, and other services. Meanwhile, we have also witnessed significant development of government intervention and policies to support farmer cooperative development toward diversification, capacity building, regular management, and complication to cooperative principles. The supporting policies include fiscal and financial subsidies, tax reductions, etc. In recent years, the Chinese government has paid more attention to the standardization and complication of cooperatives under the FPC Law through various policies and measurements, including recognition and promotion of model cooperatives, cooperative education, training, etc.

In terms of financial support for cooperative development, before the enactment of the Cooperative Law, the accumulation of government funds was only CNY 270 million GBP 1 = CNY 8.68) in the past 20 years. In 2007, when the Cooperative Law was promulgated, the figure reached CNY 220 million for one year only and then increased yearly. From 2007 to 2013, a total of CNY 9.577 billion was allocated, with an average annual support fund of CNY 1.6 billion. By October 2019, 2.2 million cooperatives had been registered across the country. From the perspective of service quality, 53% of farmer cooperatives provide value-added services such as warehousing, processing, and logistics and provide integrated production and marketing services. However, cooperative developments in rural China lack awareness regarding regulation systems and compliance with cooperative principles. This can be seen from so many faked or shell cooperatives across the country due to the following factors. First, some local governments pursued the number of cooperatives as the indicator of their performance evaluations regardless of local conditions and needs of farmers. Second, some rural entrepreneurs or local companies registered cooperatives to gain or access government financial support, tax reduction, financial insurance, land, and electricity use policies. Third, many registered cooperatives could not provide effective services due to various factors. In addition, there is a regional dispersity in cooperative development across the country. Among the top 500 farmer cooperatives recognized by Farmers' Daily in 2019, 41.8% are in the eastern region, leaving central and western regions with 29.8% and 28.4% of cooperatives, respectively.

In the new "targeted poverty alleviation" campaign since 2014, cooperative development has been listed as a key dimension and criterion for the success of government performance at the county level. The rationale of cooperative development in poor areas is closely related to but not limited to local pillars or characteristic industries (either or both agricultural or non-agricultural, e.g., rural tourism) for farmer income growth, credit cooperation, and appropriate agricultural technologies [19]. The establishment of farmer cooperatives in poor areas represents an important point of access for local governments to transfer central government funding to targeted households or communities as the share of the latter's capital investment for the initiative or development of cooperatives. Accord-

ing to official figures, more than 90% of officially recognized poor villages have set up cooperatives, with a total number of 682,000 in 832 nationally registered poor counties. Furthermore, a total of 3.851 million registered poor households have joined cooperatives with more than 29.78 million rural poor striving to achieve income growth to various extents. Similarly, poor rural households can gain other benefits, including low-cost agricultural services provided by government agencies or commercial companies, additional income from the value chain extension, and the share dividends from "their" capital investment provided by the government.

Certainly, there are some deficiencies in government-led cooperative development in China's impoverished regions. First, through strong administrative measures and financial simulations, the consolidation of "shell cooperatives" is inevitable. Second, for many local government departments, their involvement in cooperative development is not to promote local industrial development and farmer cooperation, but to ensure that government poverty alleviation funds are not misused by rural elites or private enterprises. Third, it is not surprising that in many cases poor rural households responded less well to government campaigns in new cooperative initiatives than wealthy households, leading to the development of exclusive cooperatives. Fourth, in some cases, due to the narrow definition of targeted poor households entitled to government funds or financial subsidies, cooperative development may incur additional costs, leading to divisions within rural communities.

## 4. Methodology of Research Design and Fieldwork

In response to the knowledge gaps in cooperative research and long exploration in China, this section aims to define government intervention in cooperative development as guidance for our research design and justify the case study and sample selection, and fieldwork methodology.

### 4.1. Definition and Measurement of Government Intervention for Cooperative Development

In the context of targeted poverty alleviation (TPA) and cooperative development in China, the term *government intervention* here refers to local government action at the county level that seeks to influence or change attitudes, perceptions, and behaviors of farmers, community leaders, and key stakeholders, internally and externally, for the initiation, consolidation, and dissemination of farmers' specialized cooperatives. Government intervention for cooperative development can be measured from several indicators such as political commitment, critical resources (e.g., financial subsidies and professional support), and key personnel appointed responsible for cooperative affairs. Unsurprisingly, government intervention varies greatly depending on many factors and conditions, from local resource endorsement and infrastructure relevant to industrial strategy and external markets to local government leadership. For this paper, we pay special attention to the process of cooperative development with the following questions:

- How does a local government initiate a process of cooperative development? Such questions may be related to many factors such as the motivation of local government, selection of geographic location and pillar industry, sources of householder livelihoods among local farmers, critical resources and favorable policies, etc.;
- By what mechanisms does local government coordinate multiple stakeholders to mobilize and support local farmers to participate in the "experiment and demonstration" of cooperative development? These may involve, but not be limited to mutual trust between external stakeholders and rural communities, the experience and expertise of cooperative professionals, the availability and quality of the cooperative leader assigned by the government, the timing and the access to external markets, etc.;
- By what indicators can we measure the success of government intervention? These may include, but not be limited to: the growth of cooperative memberships and number of cooperatives; the increase in the share of cooperative products/services in

external markets; the significant contribution to technological learning, householder income growth, and reduction of rural poor, etc.

The above definitions provide a guideline for research design and fieldwork described in the next two sections, as well as data analysis and presentation in Section 5.

### *4.2. Case Study and Sample Selection*

Based on the above research questions and framework for empirical research on government intervention, a case study is adopted for the following reasons.

Firstly, this is explorative research in nature as little research has been conducted on government intervention in cooperative development in poor areas of China. Secondly, the location of this project was based on the remote, mountainous, and Tibetan-ethnic-minority zone of Sichuan where cooperative development is still in its infancy. Thirdly, different from other successful cases of government intervention based on local resources and talents for cooperative development, we studied this case based on a successful introduction of an external cooperative, a new but potentially good practice in the context of "targeted poverty alleviation" to our knowledge.

We selected HS County to host our project because HS can represent poor areas of the Tibetan Zone of Sichuan in terms of its geography, resources, and rural development. More importantly, the HS government has successfully introduced an external cooperative to facilitate cooperative development in beekeeping, a local pillar industry for poverty alleviation. Regarding geographic representativeness, HS is in the mountainous canyon in the middle of the Aba Tibetan and Qiang Autonomous Prefecture in the south of the Western Sichuan Plateau, with an altitude of 1790–5286 m above sea level and an area of 4356 square kilometers. It is occupied by a population of 59,900 (2017), of which 80% are registered as rural residents and 20% as urban, and the vast majority are Tibetans (93%), with the rest comprising Han Chinese and other ethnic minorities. The annual disposable income per capita of urban and rural populations was CNY 30,000 and CNY 12,000, respectively, in 2017 [20].

Furthermore, 10,300 rural people (most of which are young people) fall into the category of migrant workers (spending six months or more each year outside of the country), accounting for 17% of the total population [20]. The cultivated land area is 66,000 hectares, cropped mainly with wheat and potatoes. The county government listed six pillar industries: early-fruiting walnut, Tibetan pig, traditional Chinese medicine, phoenix-tail chicken, ecological vegetables, and HS honey (Heishui county people's government website: http://www.heishui.gov.cn (accessed on 15 February 2023); Statistical Bulletin of 2017 National Economic and Social Development of Heishui County).

These are the reasons that we used to select Xin-Mu Beekeeping (XM Cooperative" thereafter) for this case study:

1. Beekeeping has been identified as a key poverty alleviation industry by the Sichuan Provincial Government. As the largest honey-producing province in China, Sichuan has more than 70,000 beekeepers. In 2019, the total output value of bees was 55,000 tons, with an output value of CNY 6 billion. More than 50 poverty-stricken counties recognized by the state have been identified as key honey counties. Production poverty alleviation helps more than 20,000 impoverished households to file and register.
2. HS is a typical county producing high-end honey in the Tibetan Region of Sichuan. This is because HS is a "pure land" preserving its original natural features without any industrial pollution. It is rich in plants such as wild-teeth willow, sea-buckthorn, and Gallnut, providing sources for high-quality honey. Additionally, there is a long history and experience of beekeeping in HS, shared by all villages across the county. Living at high altitudes (over 3000 m above sea level on average), local bees have strong resistance to cold and diseases;
3. Beekeeping has been listed as a pillar industry for poverty alleviation. This allowed the county government to apply and coordinate external resources and support (e.g., training and consultant services) from the provincial government and their

partners outside to expand the honey market and move bees to low-altitude regions in cold winters to cope with low honey production issues. External participation via government intervention plays a vital role in tackling barriers to cooperative development from both technical and organizational aspects, as shown in the rest of this article;

4. This is a typical case representing strong government intervention in cooperative development in a poor area of China. Recognized as "the first driving force", the government has introduced a series of policies to promote and standardize cooperative development. This is particularly true for the cooperative initiative in poor areas, including HS County, where it is very difficult or almost impossible for farmers to establish a cooperative by themselves due to many constraints regarding knowledge, skills, resources, and resource interconnections with external markets. For cooperative development in this region, it is not rare for the local government to initiate a cooperative by appointing a civil servant as a chair for the farmer cooperative;

5. Different from government intervention in other locations of poor areas of China, where cooperative development is heavily dependent upon local resources and talents, the HS government introduced a provincial star beekeeping cooperative from an advanced area of Sichuan to provide training, technological support, and management support to XM Cooperative: good practice of external social enterprise's participation in cooperative development and poverty alleviation in China.

This case study originated from a previous study on CX Beekeeping Cooperative Ltd. (hereafter referred to as CX Cooperative), a provincial star cooperative located in the Chengdu plain and a leading player in the beekeeping sector. It was unexpected that CX cooperative was involved in a poverty alleviation project to help XM Cooperative in HS county. Taking a perspective of the cooperative ecosystem from the GCRF project, the case of XM cooperative was selected to develop our understanding and evaluation of the government intervention in HS County.

### 4.3. Fieldwork Methodology

The case study started with an online meeting and second data collection first, followed by fieldwork in HS county after the end of COVID-19 in May 2019. The field research in HS County was composed of site visits (XM Cooperative, a village, and five beekeeping farmers); in-depth and semi-structured interviews (10 senior managers of cooperatives, government officials, and technicians); and a group meeting attended by 11 people from the county government agencies, cooperative leaders, beekeeping professionals, and external partner representatives from Zhejiang Province.

## 5. Narrative of the Case: Government Intervention on XM Beekeeping Cooperative

Applying the framework for government intervention, this section provides a narrative of a case of cooperative development in HS, a poor county in Sichuan Province inhabited by Tibetan minorities.

### 5.1. Cooperative Initiative: The Government Took the Lead

Like many poor counties in west China, the HS government's initiative of the XM Cooperative project was related to many factors. Firstly, it was driven by a central/provincial poverty alleviation program in which cooperative development has been listed as an important indicator for the success of the poverty alleviation project at the county level. Secondly, it was selected as one of six priority sectors for the HS County government in 2014. Thirdly, this beekeeping project could not only release the comparative advantages in terms of natural resources and traditional skills locally but also address the bottleneck, i.e., mobilizing and organizing beekeeping farmers to participate in pillar development and poverty alleviation.

For the above reasons, the HS government appointed Mr. XGJ, a civil servant in the county government, to establish and lead XM Cooperative as a chairperson in 2014.

Mr. XGJ was selected by the government not only because he is a Tibetan professional specializing in beekeeping technology but also because he had gained a reputation and trust among beekeeping farmers due to his working experience for 20 years. Thanks to good preparation and effective promotion, the project's initiative was very successful, and more than 50 farmers from 15 villages joined the cooperative, with an average of 30 bee colonies per household in the first round. The scope of the cooperative business included HS bee breeding, honey product acquisition sales, and technical information services. The number of XM Cooperative members and bee colonies increased to 96 and 5200, respectively, resulting in 78 tons of HS honey and CNY 3.9 million in sales in 2015. Furthermore, under Mr. XGJ's leadership, XM Cooperative provided a training course for core members. It started the process of the cooperative brand through participation in relevant Fairs at national and provincial levels and registration of "HS Honey" at the National Authority.

Following a reform of decoupling between administrative units and enterprises, Mr. XGJ resigned from Chair of XM Cooperative in 2016 and was replaced by Mr. RX, a local beekeeping farmer and key technician of the cooperative. Mr. RX was also a Communist Party Branch Secretary of the village (or Head of the village) with an advantage in coordinating village resources, organizing farmers, and docking government policy. All the above factors were important for him to win the trust and support of most of the cooperative members. This is a significant step for XM cooperative development from government-led to farmer-led management. Despite a successful power transfer from the government agency to cooperative members, XM Cooperative was facing many challenges. First and foremost, it was critical to offer all cooperative members training to help them change their mindset and traditional approach to beekeeping, resulting in a poor yield and tiny bee colony size. A significant difficulty was also the high hilly plateau, contributing to the poor bee survival rate over the long, harsh winter.

Second, although being registered, the HS Honey brand did a terrible job of recognizing the external market and the uniform standard of goods it utilized within. Due to the restricted and unstable size of the cooperative's product sales, the benefits of cooperatives in terms of product quality and profit return were seldom ever realized.

Third, there was not yet an effective governance system within the cooperative to link individual production and united sale. As a result, cooperative members were not obliged to sell their products through the cooperative, resulting in insufficient honey supply in the cooperative. In addition, there was a significant difference in terms of honey prices between the cooperative and individual farmers in the local honey market. For instance, the honey product of the cooperative was sold at CNY 60 per 500 g, 50% higher than the market average.

*5.2. Transformation of XM Cooperative: Partnership with CX Cooperative*

To cope with the above challenges, the HS government began to seek external support through the Beekeeping Management Station of the Agriculture Department of Sichuan Province in 2015. CX Cooperative, a national star in this sector, was recommended. CX Cooperative is a national star in the beekeeping sector, located in Qionglai, a part of Chengdu Municipal of the Provincial Capital. Mr. WS, chair of CX Cooperative, was awarded the title of "Outstanding Asian Bee Farmers" by the Asian Apicultural Association (AAA) in 2008. The cooperative was selected as one of the national Top 100 bee cooperatives by the Ministry of Agriculture in 2018. It was deemed appropriate for CX Cooperative to develop a partnership with XM Cooperative to expand its high-quality honey sources in the minority areas for a comprehensive park integrating the production and marketing of high-quality honey. The HS government brought CX Cooperative to work with XM Cooperative through a project bidding in terms of operation. Accepting the invitation of the HS government, CX Cooperative took the following measures for XM Cooperative to cope with the challenges.

The first was to establish a cooperative management system that contained profit distribution, an accumulation fund, a mutual risk reserve fund, and an endowment insurance subsidy for members. Considering the conditions of the scattered villages, high mountains, steep roads, and lack of beekeeping technicians in HS County, CX Cooperative designed a unified standard and procedure for cooperative members to follow, from the selection of bee species to the packaging and sales of products. The second was to provide bee species, bee colonies, bee machines, and technical services for XM Cooperative through a contract purchased by the HS government. Furthermore, Mr. WS, Chair of CX Cooperative, was appointed an adviser to XM Cooperative, along with five technicians from CX Cooperative staying in XM Cooperative throughout the year to provide technical guidance and support. Meanwhile, XM Cooperative members had chances to visit CX Cooperative. The third was to share its market platform and customer resources with XM Cooperative to sell XM products and expand marketing channels through e-commerce and social media (e.g., WeChat). CX Cooperative can sell about 30% of XM products each year through the above measures. The partnership with CX Cooperative has led to the rapid development of XM Cooperative. For instance, XM Cooperative has successfully applied for two patents for HS beehives and packaging and has registered one trademark of Yakexia. HS Honey was one of the first batches of regional public brand products of "Pure Land Aba" awarded by the Aba Prefecture Government. In 2017, XM Cooperative successfully obtained EU organic product certification and participated in the 15th China International Agricultural Products Fair. In 2018, XM Cooperative built 46 demonstration farms certificated by the EU organic bee apiary, and by 2019, the number had reached 66.

*5.3. Scale-up of XM Cooperative: Establishment of Cooperatives Union*

Having been a partner with CX Cooperative, XM Cooperative has made remarkable progress in technology and management, increasing the impact on other beekeeping farmers becoming members across the county. Recognizing the value of standardized management for cooperative development, the HS government encouraged and supported XM Cooperative's alliance with two other beekeeping cooperatives as a joint XY Cooperative Union in 2018, led by XM Cooperative.

The XY Cooperative Union has led to the establishment of 6 new beekeeping cooperatives with more than 700 households participating county-wide, including 13 poor villages and 380 poor households since 2019. From 2018 to 2019, XY Cooperative Union colonies grew from 2300 to 4000, its product yield increased from 11.5 to 30 tons, and its output value increased from CNY 690,000 to CNY 2.4 million, thus increasing by 173.9 percent, 260.9 percent, and 347.8 percent, respectively. In addition, 15 cooperative members were hired by the county government as "experts" to provide training and technical services for beekeeping in 22 villages.

*5.4. Market Expansion Crosses Regional Boundaries via Government Intervention*

Government intervention has not only led to a successful partnership for XM Cooperative but has also played a key role in the further expansion of HS Honey across regional boundaries. The importance of HS government involvement can be illustrated by the distribution of XM products: only 40% of sales occur in the local market, while the other 60% occur in the external market. The latter sales are related to a cross-regional collaboration under the arrangement of a national poverty alleviation policy in addition to the CX Cooperative channel.

The central government put forward the policy of east–west collaboration for rural poverty alleviation in 1996. Through administrative means, the central government arranges government agencies, state-owned enterprises, and public institutions in economically developed (mainly eastern) regions to help address the needs of an assigned prefecture or county in poverty-stricken areas of western regions. The assistance can take various forms, including financial support, technical transfer, and professional participation. To ensure the timely realization of the national target of "targeted poverty

alleviation" by 2020, the central government further strengthened the coordination between the eastern and the western parts of China to accomplish poverty alleviation in 2016. It defined cooperation in five aspects, including industrial cooperation and labor service cooperation. Under this framework, Zhejiang and Sichuan Provinces established a pair. Two municipals of Zhejiang, namely Tongxiang and Haining, have become counterpart supporters of HS County. Accordingly, the Haining government invested and established HC Investment and Development Co., Ltd. in HS in 2018 to open the external market for HS agricultural products.

Taking this opportunity, XM Cooperative has developed strategic cooperation with a famous agricultural supply chain enterprise in Haining by creating a joint venture company to sell HS Honey to high-end customers in the Yangzi River Delta market. At present, the share of XM products in this market has increased to 15% of the cooperative sales. In addition to coordinating poverty alleviation between the eastern and the western parts of China at the central government level, Sichuan provincial government has also arranged a group of counties and cities with relatively good economic foundations and strong financial strength to support poverty-stricken counties in Tibetan areas. Under this framework, HS County has opened six specialty stores of agricultural products in Pengzhou, a Municipal near Chengdu, to sell high-quality agricultural products to customers of the provincial capital. Currently, this market shares another 15% of XM products.

## 6. Discussion

The complexity and new momentum of cooperative development in poor areas of China offer a unique opportunity to conduct empirical research on successful conditions of government intervention. By linking research questions at the beginning of this paper with working definitions of government intervention in Section 4, this section aims to summarize research findings from the XM Cooperative case study and draw implications for cooperative development and external intervention in poor areas of rural China and beyond.

### 6.1. Understanding Government Intervention from the Lens of Cooperative Development

In our view, there is a knowledge gap in cooperative research regarding the lack of an in-depth understanding of the "cooperative development context" [5], which is crucial for cooperative development in poor areas of the developing world. Given the complexity and diversity of local environments, resource endorsements, development needs, and cooperation culture, we argue that external (including government) intervention plays a vital role in initiating a process of cooperative development to empower smallholder farmers to overcome multiple constraints or "poverty traps" [21]. Viewing local government intervention as an important element of a cooperative ecosystem [3], many implications can be drawn from our research findings.

Firstly, cooperative development in rural China is not an end in itself, but an important means used by local governments to promote intervention for local development, poverty alleviation, and farmer empowerment. This is particularly true in poor areas of China due to two reasons: (1) cooperative development was serviced for local economic growth through the selected pillar of industrial development [22]; and (2) compared to developed areas of China, there were many constraints, especially regarding talent, civil society organizations rural entrepreneurship, and social innovation, including the emergence of these elements from cooperatives.

Secondly, governmental intervention is vital in constructing a cooperative ecosystem to bring relevant elements or conditions together to initiate and facilitate cooperative development. This can be seen from the rapid expansion of cooperatives across rural China since the Law of Farmer Professional Cooperatives was put into effect in 2007. The government-led cooperative ecosystem is particularly important in poor areas where the variety of cooperative development is primarily attributed to the differences between local governments in terms of approaches, commitments, and economic development strategies.

Thirdly, the role of government intervention cannot be overemphasized, as cooperative development in poor areas is determined by many factors within the ecosystem, of which government intervention is only one. Nonetheless, governmental intervention can influence or reshape the cooperative ecosystem according to the feedback from the success or failure of the intervention.

In short, from the perspective of the cooperative development context, this case offers insight into the implementation of the national program "Targeted Poverty Alleviation" at the local level in general and various roles or performances of government intervention in cooperative development in poor areas of China in particular. This case provides a reference for further research and debate about the role of government intervention for cooperative development in poor areas of the Global South, which may conflict with principles of cooperatives drawn from the Western world [4].

*6.2. Measuring Government Intervention from Multiple Aspects*

Bearing in mind the nature of exploring research in the context of TPA, stronger government intervention provided a unique opportunity to design and conduct a case study for government intervention in poor areas of China. Several observations can be drawn from this case.

Firstly, government intervention at the local level can be observed from multidimensional inputs: political commitment for cooperative development; local pillar industrial (or product-related) strategy; appropriate technology; and market connection and organizational involvement. Furthermore, the roles and tasks above are related to different sources of resources, the hierarchy of the political system, and job division within government agencies. The different outcomes of government intervention can be attributed to the strength of individual elements and, more importantly, the interconnection and coordination between them. In this regard, the failure of government intervention can be found in the absence of the above elements or the mismatch between them.

Secondly, government intervention in cooperative development as a top-down process cannot be successful until the intrinsic dynamics within the rural community are identified, developed, and interface with external resources and the market. This can be used to explain the necessity of government intervention in poor areas to initiate a process of local development, including cooperative development through an "experiment, test, and demonstration" approach, and the change in government intervention in terms of strengths and styles according to the stages of cooperative development. More importantly, we argue that identifying the interfaces between intrinsic dynamics and external resources/opportunities can lead to similar outcomes as those achieved in bottom-up development [23–26].

Thirdly, the success or failure of government intervention can be measured through the outcomes of cooperative development, which contain at least three interwoven dimensions: market expansion of local products or services; technological advances; and diffusion and organizational consolidation. Such outcomes, however, cannot be attributed to government resources or inputs themselves but result from all the resources and factors surrounding them, in which local government plays an important role in facilitating and coordinating multiple stakeholders, internally and externally, in poor areas. This finding coheres with a conclusion from other research on government intervention for farmer innovation diffusion [25,26].

*6.3. Conditions of Successful Government Intervention in Poor Areas*

Given the complexity of cooperative development and government intervention in poor areas of China, this paper examines the case of XM Cooperative to illustrate the conditions for successful government intervention for cooperative development. Several research findings can be summarized from this case.

Firstly, this case shows many conditions necessary for successful intervention including (1) appropriate selection of a local pillar industry based on local resource endorsements

and householder livelihood systems for cooperative development; (2) an interface with local technologies and an external market; and (3) organizational support, especially selection, training, and appointing key persons to lead a cooperative for "experiment and demonstration". In the case of XM cooperative, the governmental intervention met the first two conditions due to its unique natural environment and distinguished advantage in high-quality honeybee resources as well as its interface with local native beekeeping traditions. Given the constraints on talents for cooperative leadership in poor areas, it is not rare that the HS government appointed a civil servant who was familiar with the local community and who had expertise in relevant technologies as the first chair of XM Cooperative: an important condition for the successful initiative of the first cooperative in that county.

Secondly, it may be a long way to go for a successful governmental intervention until cooperative leadership emerges among participatory members, associated with establishing and effectively operating a management system. With a focus on identifying and releasing intrinsic dynamics within rural communities, the most impressive and effective measure taken by a local government, in this case, is the introduction and participation of a successful beekeeping cooperative from developed areas. This could ensure the participation, training, and technological services of cooperative members and accelerate the establishment of a cooperative management system internally as well as open partnerships with external collaborators for the sale of local honey products.

Thirdly, the successful establishment and maintenance of a sponsored cooperative are necessary but not sufficient for a successful intervention. This is because government intervention plays an important role in facilitating the diffusion of both technological and organizational innovation for sustainable rural development in the wider community and not just in poor areas of the developing world [25,27]. In addition to the introduction of CX Cooperative from the outside, the county government contributed to the XM Cooperative's development through the following measures: (1) it supported the testing and registration of local honey as an organic product under EU food standards; (2) it helped XM Cooperative establish a partnership with an agricultural supply company in the coastal areas to develop the high-quality honey market in the developed region of China; and (3) it facilitated the collaboration and emergence of beekeeping cooperatives to share technology, management, and market resources with XM Cooperative. All the above measures contributed to a significant improvement of the cooperative ecosystem in HS county, leading to sustainable cooperative development in all areas from market expansion and technology diffusion to organizational consolidation.

Fourthly, government intervention in poor areas varies with cooperative development, which can be further divided into three stages according to the XM Cooperative case:

- Initiative: selecting pillar industrial areas for cooperative development; identifying the needs of technology training and interfaces with the external market; providing organizational support for screening and appointing proper cooperative leaders;
- Facilitation: attracting the participation of local farmers, the exploration of appropriate technology and external markets, leadership training, and the development of social capital (trust, norms, and networking internally and externally);
- Consolidation: focusing on improving regulation and management systems, partnerships for external markets, and long-term cooperative development, scaling up and practicing innovation diffusion for more participation and to bring benefits to local farmers and the rural poor in particular.

### 6.4. Policy Implications for Government Intervention in Cooperative Development

Bearing in mind many failed or ineffective government interventions, in reality, this case has many policy implications for government intervention in poor areas of China.

- A rigorous procedure for preparing and discussing government intervention in cooperative development is needed to ensure the participation and coordination of all

stakeholders, both internally and externally. This could lead to a significant decline in the unsuccessful rate of government-led projects;

- It could be more effective and efficient if an external cooperative or relevant NGO is invited as a key stakeholder joining the discourse surrounding potential government intervention to ensure the identification, mobilization, training, and empowerment of cooperative leadership and core members on the one hand and to develop cooperative cultural norms on the other;
- The central focus of government intervention should be on creating and facilitating a favorable platform rather than on the quantitative growth of cooperatives or on narrowly selecting "modeling cooperatives" that are difficult to duplicate or sustain without strong support from the government. This is because governmental intervention will not be successful until the emergence of cooperative leadership from rural communities leads to sustainable cooperative development;
- In the context of TPA in poor areas of China, government intervention for cooperative development varies with geographic location, local pillar industries, economic development, andsocial–cultural environments. A systematic collection, analysis, and comparison of cooperative development in poor areas are necessary for better understanding, planning, monitoring, and evaluating a governmental intervention for cooperative development in different regions and programs including ongoing rural revitalization campaigns.

## 7. Conclusions

Strong government intervention in the context of "targeted poverty alleviation" (TPA) in poor areas of China offered a unique opportunity to conduct empirical research to understand the rationale and conditions of successful government intervention to initiate and facilitate cooperation development in this region. Based on a case study of XM Beekeeping Cooperative in marginal areas of Sichuan, several conclusions can be drawn from this paper. Firstly, government intervention at the county level can play a vital role in coordinating multiple resources and opportunities, both internally and externally, to initiate a process of cooperative development in poor areas. Secondly, successful government intervention depends upon many interwoven factors and conditions, of which political will, interfaces with local resources and pillar industry strategies, and cooperative leadership are most important. Thirdly, top-down government intervention will not be successful unless intrinsic dynamics and cooperative leadership in local communities are fully identified and properly trained; in this regard, external cooperative expertise or NGOs can play a positive role. Finally, this case illustrates the importance of developing a "cooperative development context" to understand local challenges and good practices in poor areas of the Global South on the one hand and the role and boundaries of government intervention for cooperative development on the other.

**Author Contributions:** B.W. provided the concept, theme, and framework for empirical study in Sichuan and drafted all of the manuscripts except Sections 3 and 5. S.Z., R.C. and J.L. developed and outlined this idea, including the method and approach to be used; S.Z.; S.Z. and B.W. contributed to the methodology and revision of this manuscript; S.Z., B.W. and R.C. wrote the article. N.K. and R.L.R. revised the manuscript. All authors have read and agreed to the published version of the manuscript.

**Funding:** University of Nottingham Global Challenges Research Fund (UoN-GCRF) sponsored project: Cooperative ecosystem to empower small farmers in the poor areas of China (RIS 2427898/2180292).

**Institutional Review Board Statement:** Not applicable.

**Informed Consent Statement:** Not applicable.

**Data Availability Statement:** Not applicable.

**Conflicts of Interest:** The authors declare that they have no conflicts of interest.

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
