# Peer review of "Government Intervention on Cooperative Development in Poor Areas of Rural China: A Case Study of XM Beekeeping Cooperative in Sichuan"

_land, doi:10.3390/land12040731_

Round 1
Reviewer 1 Report
In the Abstract. Please write more exactly: what was the aim of your research, main methods, and main suggestions. You say: "New light on government intervention" (1 p.) it is not clear.
-Need to present the source of the information? (8 p.)
-The article can be shorter. Some ideas (about government influence on cooperatives) are repeated.
Methodology. What were the main questions and conclusions of this interview? (9 p.)
-Must be separate Conclusions (in shorter form from the previous Discussions) (14 p.) Now it is not bad, but must be shorter and more clearly, and more concentrated.
- Some suggestions are in the text of Article.

Author Response
Comments and Suggestions for Authors
In the Abstract. Please write more exactly: what was the aim of your research, main methods, and main suggestions. You say: "New light on government intervention" (1 p.) it is not clear.
Answer: Thank you. We have modified the abstract section accordingly.
-Need to present the source of the information? (8 p.)
Answer: We have added a suitable link.
-The article can be shorter. Some ideas (about government influence on cooperatives) are repeated.
Answer: We have revised the complete manuscript accordingly.
Methodology. What were the main questions and conclusions of this interview? (9 p.)
Answer: Thank you. We have revised the methodology section.
-Must be separate Conclusions (in shorter form from the previous Discussions) (14 p.) Now it is not bad, but must be shorter and more clearly, and more concentrated.
Answer: Revised accordingly. Please refer to both sections.
- Some suggestions are in the text of the Article.
Answer: Okay. Thank you.

Reviewer 2 Report
This paper starts from the premise that there is a gap in the cooperative literature, which fails to recognize the positive role of governments for cooperative development, and then goes on to present and discuss a case study (that of XM cooperative in XS county of Sichuan, China). The latter substantiates the claim there is indeed a very important role of government in key stages of cooperative development, from creating the conditions at the start, supporting the identification and utilization of marketing channels beyond the immediate locale, and so on. The paper is well written and the main argument is at first impression appealing, especially as it draws attention to contextual factors in poor rural regions in the developing world, which require a different approach to cooperative development. However, the paper fails to develop a convincing story, as it suffers from a narrow view of the literature, does not propose a real study methodology, and makes largely untested claims.
Starting with a review of the literature, there are actually many studies which have debated top-down versus bottom-up approach to agricultural cooperatives (for instance, in the case of Ethiopia), which are not represented in the paper.
The methodology is a black box where hypotheses are not explicitly formulated, and alternative interpretations are not contemplated. The authors collected data from various individuals and informants; but it is not clear how the data were analyzed, and which of these voices contribute to which piece of the case study narrative. One would expect to hear multiple, overlapping and sometimes discordant voices as well as interests: instead, the case study is laid out as an exceptional case of uniform support and consensus, without any inflexion.
The purposes of a cooperative are manifold and certainly context dependent – but one could hardly question that a main feature of a successful cooperative is that it provides benefits to members they would not otherwise attain. However, we are never told whether members of the XM cooperative are better off, either economically, socially, or else, as effects of being part of it. The authors do not provide an initial definition of what constitutes success for a cooperative from the viewpoint of its members. By not defining success through internal criteria, they conclude their study by stating that “cooperative development in the developing world is not an end by itself, but a means of government intervention for local development and poverty alleviation” (551-2) and that “governmental intervention is vital in constructing a cooperative ecosystem” (558). At the end, the reasoning becomes circular and the reader remains unconvinced.
Minor questions: What is XM?
Author Response
Comments and Suggestions for Authors
This paper starts from the premise that there is a gap in the cooperative literature, which fails to recognize the positive role of governments for cooperative development, and then goes on to present and discuss a case study (that of XM cooperative in XS county of Sichuan, China). The latter substantiates the claim there is indeed a very important role of government in key stages of cooperative development, from creating the conditions at the start, supporting the identification and utilization of marketing channels beyond the immediate locale, and so on.
Answer: Thank you for your comments and suggestions. We have revised the manuscript accordingly.
The paper is well written, and the main argument is at first impression appealing, especially as it draws attention to contextual factors in poor rural regions in the developing world, which require a different approach to cooperative development. However, the paper fails to develop a convincing story, as it suffers from a narrow view of the literature, does not propose a real study methodology, and makes largely untested claims.
Answer: According to your comments and suggestions. We have revised and restructured all sections accordingly. Thank you.
Starting with a review of the literature, there are many studies which have discussed top-down versus bottom-up approach to agricultural cooperatives (for instance, in the case of Ethiopia), which are not represented in the paper.
Answer: Thank you for your great suggestions. We modified the literature section focusing on Top-Down and Bottom-up approaches.
The methodology is a black box where hypotheses are not explicitly formulated, and alternative interpretations are not contemplated. The authors collected data from various individuals and informants, but it is not clear how the data were analyzed, and which of these voices contributed to which piece of the case study narrative. One would expect to hear multiple, overlapping, and sometimes discordant voices as well as interests: instead, the case study is laid out as an exceptional case of uniform support and consensus, without any inflection.
Answer: We have revised all methodology of research design and fieldwork accordingly.
The purposes of a cooperative are manifold and certainly context dependent – but one could hardly question that a main feature of a successful cooperative is that it provides benefits to members they would not otherwise attain. However, we are never told whether members of the XM cooperative are better off, either economically, socially, or else, as the effect of being part of it. The authors do not provide an initial definition of what constitutes success for a cooperative from the viewpoint of its members. By not defining success through internal criteria, they conclude their study by stating that “cooperative development in the developing world is not an end by itself, but a means of government intervention for local development and poverty alleviation” (551-2) and that “governmental intervention is vital in constructing a cooperative ecosystem” (558). At the end, the reasoning becomes circular and the reader remains unconvinced.
Answer: Thank you for your important suggestion according to government intervention for local development and poverty alleviation. We have revised and removed the unnecessary sentences and world such as poverty alleviation.
Minor questions: What is XM?
Answer: Xing-Mu. It is defined in the introduction section's second last paragraph.

Reviewer 3 Report
Please, read atached file

Author Response
Line 146- 150 = The references of Fals et al and Gagon (1970-71) and Develtere (2009) correspond to frameworks very different. Therefore, it must be considered in the analysis
Answer: Thank you. Revised accordingly.
Line 146- 150 = Reference s10 and 8 must be revised
Answer: Corrected. We have removed ref 8.
Line 146- 150 = This paragraph is very interesting but there is no reference/source
Answer: Corrected.
Line 559- 568= The paragraph numbers are wrong
Answer: Corrected.
441- 445= this last idea doesn’t make sense
Answer: Revised. Thank you
- Comments and suggestions
Mains contributions:
1) Awareness of two interrelated factors:
- the difficulties of setting a universal pathway for cooperative development
(lL544 to 546).
Answer: Thank you. We have revised the complete section and also separated the “ discussion and conclusion sections
- the importance of knowing each one of the ‘’ cooperative development context’’ L 542)
Answer: We have improved the recommended section. Please refer to the discussion section
2) Detecting the main problem (L 156-157) : in a rapid increase of cooperative membership without attention paid to the lack of cooperation spirit among rural people from the beginning to: genius of the cooperative movement is the strength of its grass-roots support so that whether the government intervention "makes little difference in the long run" because many co-ops thrive anyway and become independent of outside support. This lack is linked to (L265- Third, it is unsurprising that in many cases, the response to the government campaign in a new initiative of cooperatives is lower among rural poor than wealthy householders, resulting in an exclusive cooperative development.
Answer: Thank you. We completely agree with you. However, we also discussed that it is not necessary if one approach works in one country; it will perfectly work in other countries. Lines 156-156 are in the context of India, while we are talking about a particularly poor region of China.
Main problems
1) This last subject is linked with main problem of the paper: Results are focused int the entrepreneurial success of the cooperative, but there are no indicators about poverty alleviation, maybe income growing, or others. So, this idea (L551): Firstly, cooperative development in the developing world is not an end by itself, but a means of government intervention for local development and poverty alleviation "can’t be tested with the research"
Answer: Corrected. Please refer to the title and subtitle of the discussion section (6.1-6.4).
2) Sections 4 and 5 are wrong formulated.
-Section 4 Research design and evaluation framework for government intervention. There is an inconsistency between this title and its matter:
Answer: Corrected. We have changed the sections 4 and 5 titles and subtitles.
Answer: Corrected.
Answer: Thank you. We have modified it accordingly.
- c) L 391-398 enunciate field research, whose results are not included-quoted in any section.
Answer: We have removed unnecessary sentences. Please check section 5. Thank you.
3) As the authors recognize in section 6.5., the suitability/representativeness of beekeeping sector regarding to the rural cooperatives of China is low, because the ratio of land ownership is low.
Answer: We have removed section 6.5 and added a separate conclusion section
4) Most of the analysis focuses on government intervention. For example, L 499-531 are really referred to the government intervention content (normative/program), not to their issues regarding to poverty alleviation.
Answer: Thank you for your suggestions. Particularly in this case study, we focused on one of the poverty alleviation tools: Beekeeping. In addition, government intervention is an important tool for poverty alleviation. In the new campaign of “targeted poverty alleviation” since 2014, cooperative development has been listed as a key dimension and criterion for the success of government performance at the county level. Therefore, we did not explore and discuss all other tools and approaches for poverty alleviation so that we stay focused on one particular approach for this case study.

Round 2
Reviewer 2 Report
I would like to congratulate the authors for having thoroughly revised and refocussed the manuscript. I have enjoyed reading this new version of the paper, which to me now provides an interesting contribution to assess the role of government in supporting cooperative development in rural, poor China. The authors offer very interesting insights on how top-down development can incorporate local dynamics and innovations to generate interesting institutional and market outcomes. The way that evidence gathered during fieldwork was used in the analysis ("who said what") is not totally transparent, but I take this as case study more than qualitative analysis.
The paper is generally well expressed but there are multiple instances where expression is not clear, or is grammatically odd. I have taken quite a bit of time indicating the parts that require attention, and are listed below (per line number). Please revise them all:
207: I think you meant “Southern Africa” (which Tanzania is part of) not “South Africa”
207: “the latter” instead of “later”
216: delete “whether”
270: what do you mean by “far too mature in regulation systems and compliance”? please rephrase more clearly
280: substitute with: “with 29.8% and 28.4% of cooperatives, respectively”
287: “represents an” rather than “is serviced as”; or clarify otherwise.
293: “striving to achieve” rather than “for their”
298-309: some of these are not really deficits but unintended effects (the second for instance): can you rephrase or explain them better, especially the first?
312: “In response” rather than “In reflection”
316: “and justify the” rather than “justification of”
321: spelling of Definition
327: “initiation” rather than “initial”
405: add: still “in its” infancy
407: add “was” based
408: add: a new but “potentially” good
525: substitute “were” for “are”
554: “it was deemed appropriate” rather than “it was right”, or some similar expression
586: what does it mean “increasing the impact on other beekeeping farmers”? that other farmers imitated or became members?
588: “with two other beekeeping cooperatives” because I am not sure you identified those two
609: correct into: “(mainly eastern) regions”
645: “is the lack of” rather than “lacks”; or similar edits
656: write: “implications can be drawn from the research findings” or similar
681: insights “into”
690-93: this sentence provides a circular argument and I am not sure it is necessary. I would go straight to the point below with a brief introduction
704: “the lack of one” rather than “the missing”
711: “and” instead of “also”
715: “can lead to similar outcomes as those from” rather than “leads to”
716: “through” rather than “from”
722: “in which” rather than “to which”
726: “conclusions” rather than “with a conclusion”
817: “focus on” rather than “be paid to”
820: “without a strong support from…”
834: “evaluating” rather than “evaluation of”
844: “can play” rather than “plays”
Author Response
Comments and Suggestions for Authors
I would like to congratulate the authors for having thoroughly revised and refocussed the manuscript. I have enjoyed reading this new version of the paper, which to me now provides an interesting contribution to assess the role of government in supporting cooperative development in rural, poor China. The authors offer very interesting insights on how top-down development can incorporate local dynamics and innovations to generate interesting institutional and market outcomes. The way that evidence gathered during fieldwork was used in the analysis ("who said what") is not totally transparent, but I take this as case study more than qualitative analysis.
The paper is generally well expressed but there are multiple instances where expression is not clear or is grammatically odd. I have taken quite a bit of time indicating the parts that require attention and are listed below (per line number). Please revise them all:
Thank you for your comments and suggestions. We have revised the manuscript accordingly.
207: I think you meant “Southern Africa” (which Tanzania is part of) not “South Africa” Corrected.
207: “the latter” instead of “later” Corrected.
216: delete “whether” Deleted.
270: what do you mean by “far too mature in regulation systems and compliance”? please rephrase more clearly Cleared.
280: substitute with: “with 29.8% and 28.4% of cooperatives, respectively” Corrected.
287: “represents an” rather than “is serviced as”; or clarify otherwise. Corrected.
293: “striving to achieve” rather than “for their” Corrected.
298-309: some of these are not really deficits but unintended effects (the second for instance): can you rephrase or explain them better, especially the first? Corrected.
312: “In response” rather than “In reflection” Corrected.
316: “and justify the” rather than “justification of” Corrected.
321: spelling of Definition Corrected.
327: “initiation” rather than “initial” Corrected.
405: add: still “in its” infancy Added.
407: add “was” based Added.
408: add: a new but “potentially” good Added.
525: substitute “were” for “are” Corrected.
554: “it was deemed appropriate” rather than “it was right”, or some similar expression Corrected.
586: what does it mean “increasing the impact on other beekeeping farmers”? that other farmers imitated or became members? Corrected.
588: “with two other beekeeping cooperatives” because I am not sure you identified those two Corrected.
609: correct into: “(mainly eastern) regions” Corrected.
645: “is the lack of” rather than “lacks”; or similar edits Corrected.
656: write: “implications can be drawn from the research findings” or similar Corrected.
681: insights “into” Corrected.
690-93: this sentence provides a circular argument and I am not sure it is necessary. I would go straight to the point below with a brief introduction Yes.
704: “the lack of one” rather than “the missing” Corrected.
711: “and” instead of “also” Corrected.
715: “can lead to similar outcomes as those from” rather than “leads to” Corrected.
716: “through” rather than “from” Corrected.
722: “in which” rather than “to which” Corrected.
726: “conclusions” rather than “with a conclusion” Corrected.
817: “focus on” rather than “be paid to” Corrected.
820: “without a strong support from…” Corrected.
834: “evaluating” rather than “evaluation of” Corrected.
844: “can play” rather than “plays” Corrected.

Reviewer 3 Report
I agree with authors' revision, but L 441-445 haven't been revised (the new text is the same)
Author Response
Comments and Suggestions for Authors
I agree with authors' revision, but L 441-445 haven't been revised (the new text is the same)
Thank you for your comments and suggestions. We have revised the manuscript accordingly.
